# Mechanical Properties and Deformation Behavior of Superhard Lightweight Nanocrystalline Ceramics

**DOI:** 10.3390/nano12183228

**Published:** 2022-09-16

**Authors:** Byeongyun Jeong, Simanta Lahkar, Qi An, Kolan Madhav Reddy

**Affiliations:** 1School of Materials Science and Engineering, Shanghai Jiao Tong University, Shanghai 200240, China; 2Department of Materials Engineering, Indian Institute of Technology Gandhinagar, Gandhinagar 382355, India; 3Department of Materials Science and Engineering, Iowa State University, Ames, IA 50011, USA

**Keywords:** boron carbide, silicon carbide, interface, grain boundary sliding, amorphization

## Abstract

Lightweight polycrystalline ceramics possess promising physical, chemical, and mechanical properties, which can be used in a variety of important structural applications. However, these ceramics with coarse-grained structures are brittle and have low fracture toughness due to their rigid covalent bonding (more often consisting of high-angle grain boundaries) that can cause catastrophic failures. Nanocrystalline ceramics with soft interface phases or disordered structures at grain boundaries have been demonstrated to enhance their mechanical properties, such as strength, toughness, and ductility, significantly. In this review, the underlying deformation mechanisms that are contributing to the enhanced mechanical properties of superhard nanocrystalline ceramics, particularly in boron carbide and silicon carbide, are elucidated using state-of-the-art transmission electron microscopy and first-principles simulations. The observations on these superhard ceramics revealed that grain boundary sliding induced amorphization can effectively accommodate local deformation, leading to an outstanding combination of mechanical properties.

## 1. Introduction

Superhard materials with Vickers hardness typically above 40 GPa are currently limited to a few materials such as diamond and cubic boron nitride (cBN) [1,2]. Of these, cBN is a representative superhard ceramic, with a Vickers hardness of 33–45 GPa [3,4]. The synthesized cBN under high pressure and high temperature (HPHT) conditions [5] showed a high hardness of 85 GPa and a high fracture toughness of 10.5 MPa.m^1/2^. Moreover, when the nanostructure of cBN was dominated by nanosized (~3.8 nm) twin domains [6], the hardness reached 95–108 GPa, which is even higher than that of synthetic diamond (hardness around 100 GPa) [7], with a high fracture toughness of 12.7 MPa.m^1/2^. These exceptional properties of superhard ceramics have led to wide industrial applications in cutting, grinding, and drilling. Following this trajectory, different studies have been conducted to find other superhard materials, and recently, hard ceramics such as boron carbide (B_4_C) and cubic silicon carbide (β-SiC), among different polymorphs of silicon carbide, have gained attention. Commercial B_4_C [8] and SiC [9,10,11] have hardnesses of 25–30 GPa and 20–27 GPa, respectively, making them insufficient to exceed the superhard threshold (>40 GPa). However, B_4_C [12] and β-SiC [13] in their thin-film forms have been reported to have hardness over 50 GPa, allowing them to achieve a superhard state.

Polycrystalline ceramics such as B_4_C and β-SiC have significant applicability due to their distinctive structural properties. The theoretical densities of rhombohedral B_4_C and β-SiC are 2.52 g/cm^3^ [14] and 3.21 g/cm^3^ [15], respectively, making them much lighter than most metals and alloys. Their excellent properties, such as high hardness, high strength, wear resistance, and chemical stability, are essential in engineering applications such as bullet-proof vests, nuclear absorbents, grinding, and polishing media [14,16]. Even with these appealing properties, polycrystalline ceramics with a coarse-grained microstructure are restricted by their innate failure susceptibility due to their brittleness, which is characterized by low fracture toughness occurring due to transgranular (also referred to as ‘intragranular’) fracture mode and rigid covalent bonding [17,18,19,20,21,22,23]. These transgranular amorphized shear bands in B_4_C have been identified with cleavage in fragmentation under high pressures or extreme impact rates [19]. The amorphous bands showed preferential alignment along certain crystallographic planes, leading to a decrease in overall shear strength under high pressures. The responsible mechanism for the formation of amorphization in microcrystalline B_4_C preferentially along the $\left(01\bar{1}\bar{1}\right)$ crystallographic plane was revealed through the disassembly of icosahedra led by shear stress [18]. Moreover, the formation of shear amorphization mediated by dislocations in single-crystal B_4_C was observed through the pop-in (sudden transition from elastic to plastic at low loads prior to crack initiation) phenomenon using depth-sensitive nanoindentation [24]. These observations confirmed that the dislocation formation mediates amorphization, rather than direct crystal-to-amorphous transition, which is usually suggested to be more critical for the failure in superhard covalently bonded ceramics. Similarly, the amorphization in β-SiC by pop-in phenomenon using nanoindentation was confirmed to cause a decrease in hardness as compared to crystalline β-SiC through molecular dynamics (MD) simulation [25,26]. This pop-in effect exhibited reversible behavior in the elastic regime, and in the plastic regime, amorphization was induced by the onset of bond angle broadening due to dislocations and dislocation loops. Owing to these failure mechanisms in polycrystalline ceramics, a reduction in grain size to a nano-regime, which is expected to enhance the combination of hardness (or strength), toughness, and ductility, provides a promising nanomechanical research direction compared to counterpart micro-sized polycrystalline ceramics [27,28,29].

Nanocrystalline materials are categorized as having average grain sizes below 100 nm with a larger volume fraction of grain boundaries (GBs), leading to significant changes in their mechanical, physical, and chemical properties [30,31]. According to the well-established Hall–Petch relationship [29,32], an increase in strength and hardness of materials is achieved by a decrease in grain size, which is shown by Equation (1):(1)$$H=H_{0}+\frac{k}{\sqrt{d}}\;,$$
where *H* indicates the hardness of a material, *H_0_* and *k* are material constants, and d indicates the average grain size of a material. Conventionally known as the Hall–Petch relationship in metals, as well as in ceramics, it is well-established above a grain size of ~μm. However, this relationship breaks down typically when the grain size becomes lower than the corresponding critical grain size of the Hall–Petch relation. An increase in the GB volume with a reduction in average grain size limits the availability of dislocation sources and pile-up within the grains [33]. Thus, a decrease in the grain size below the critical value leads to a decrease in the strength and hardness of materials, which is known as the inverse Hall–Petch relationship (also referred to as the ‘reverse Hall–Petch relationship’). Several experimental [34] and computational results [35,36] of nanocrystalline metals have confirmed the decrease in the hardness with decreasing grain size in different materials. The main attribution to the decrease in the hardness arises from dislocation activities accompanied by secondary mechanisms such as GB sliding, GB migration, and grain rotation. Accordingly, dislocation-mediated deformation observed in the Hall–Petch region translates into GB-mediated deformation in the inverse Hall–Petch region [31]. Similar to what is observed in nanocrystalline metals, the inverse Hall–Petch relationship in nanocrystalline ceramics, i.e., in MgAl_2_O_4,_ was demonstrated below 18.4 nm of grain size by the decrease in hardness due to the combined effect of conventional dislocations and GB-mediated plasticities such as GB sliding and grain rotations [37].

GB sliding, a responsible mechanism in the inverse Hall–Petch region, was proposed to enhance its mechanical properties such as plasticity, creep fracture, and ductility in different materials such as metallic alloys [38,39], germanates [40], and olivine-rich rocks [41]. Large ductility in olivine-rich rocks was observed from GB sliding induced by amorphization at GBs under high local stress [41]. The proposed understanding was suggested to be extendable to hard ceramics under high-stress conditions. Accordingly, GB sliding initiated upon oscillatory pressures in alumina (Al_2_O_3_) suggested an increased hardness and bending strength [42]. Different kinds of research on lightweight nanocrystalline ceramics have been rapidly increasing, following a trajectory of understanding the deformation mechanisms at the atomic level to enhance their mechanical properties. In this review, the nanoscale deformation process in B_4_C and β-SiC, particularly in the range near or below the crossover region (transition between Hall–Petch and inverse Hall–Petch region), are highlighted using state-of-the-art transmission electron microscopy (TEM) characterization and quantum mechanics (QM) simulations.

## 2. Effect of Grain Size on the Mechanical Properties of B_4_C and Sic

Apart from the effects of grain size itself, the mechanical properties of B_4_C and SiC also show high dependency on their properties such as composition variation [43], anisotropy [44,45,46], density [47,48,49], etc. The composition of carbon in B_4_C varies from 9 to 20%, forming B_10_C to B_4_C, among which B_4_C has superior mechanical properties [43]. Moreover, B_4_C in its single crystal form is known to be highly anisotropic [44,45]. Both hardness and elastic modulus of $\left(10\bar{1}1\right)$ crystal orientation showed higher values than that of (0001) crystal orientation. Similarly, β-SiC single crystals showed up to 44% of the anisotropic variation in Young’s modulus for (100), (110), and (111) crystal orientations [46]. Furthermore, theoretical densities of B_4_C and SiC affect their mechanical properties such as hardness [48,50], fracture toughness [48,50], elastic [47] and shear modulus [49], etc. Hence, although the densification of B_4_C and β-SiC is difficult due to grain growth during the sintering process [51,52] and its innate low self-diffusivity [53], obtaining dense (above 90% of theoretical density) B_4_C and β-SiC is important to measure their mechanical properties accurately. Therefore, to analyze the mechanical properties in and near the Hall–Petch and inverse Hall–Petch relationship, existing experimental results on the hardness and fracture toughness of B_4_C [8,50,54,55,56,57,58,59,60,61,62,63,64,65,66] and β-SiC [13,48,52,67,68,69,70,71] with a theoretical density over 90% were collected to plot grain size dependence curve in Figure 1a–d. To analyze their pure intrinsic mechanical properties, results of fabricated B_4_C and β-SiC without use of any sintering additives were used to plot Figure 1.

The conventional Hall–Petch region is observed after the plateau region (a region where grain size does not affect the hardness), starting from 9 μm to 0.43 μm of average grain size, with an increase in hardness up to 43 GPa in Figure 1a. Similarly, hardness increases from 31 GPa to 40 GPa were observed through the elimination of porosity in B_4_C by minimum solid area models, which resulted in a close fit to the Hall–Petch relationship [47]. Further decreases in the grain size result in a decrease in hardness, showing an inverse Hall–Petch relationship, whereas fracture toughness remained independent, varying between 2 and 5 MPa.m^1/2^, as shown in Figure 1a,b. Similarly, independence of fracture toughness at around 2 MPa.m^1/2^ with grain sizes of 17 μm to 120 nm was observed due to lack of toughening, which is attributed to the transgranular fracture mode [50]. On the other hand, when the density of B_4_C was above 99%, the fracture toughness was inversely proportional to grain size, indicating a greater tendency of small grains to deflect the crack path through GBs, thus increasing toughness [60].

Similar to what was observed in B_4_C, a decrease in grain size of β-SiC to 11 nm of grain size increased hardness (conventional Hall–Petch region) up to 56 GPa, while fracture toughness varied from 3 to 4.5 MPa.m^1/2^, as shown in Figure 1c–d. However, below a grain size of 11 nm in Figure 1c, an inverse Hall–Petch region is observed, showing a decrease in hardness with a decrease in grain size. Similarly, the inverse Hall–Petch region was observed in MD simulation, showing an increase in hardness with an increase in grain size by an increase in the fraction of Si-C bonds in nanocrystalline β-SiC [68]. Additionally, increased dislocation activities were observed in a single- and microcrystalline SiC than in nanocrystalline SiC, where GBs act as absorbers and emitters of dislocations.

Higher critical grain size is observed for B_4_C (430 nm) than β-SiC (11 nm) due to the poor densification led by porosity and grain growth during the sintering process. β-SiC, on the other hand, can be synthesized under HPHT conditions, suppressing the grain growth and atomic diffusion resulting in a high hardness of ~41.5 GPa with a fracture toughness of 4.6 MPa.m^1/2^ [71]. Several studies to synthesize B_4_C under HPHT conditions [72,73,74] have been carried out, although no experimental analysis exists on mechanical properties or its deformation. Thus, these factors make the narrow range of its superhard state (0.37–0.5 μm in Figure 1a) and the critical grain size of B_4_C that was observed to be much higher than the grain sizes categorized for nanocrystalline materials. In the following section, mechanical properties in the crossover region from microcrystalline to nanocrystalline, and mechanical properties of nanocrystalline B_4_C (n- B_4_C) and nanocrystalline β-SiC (n-SiC) will be discussed.

### 2.1. Mechanical Properties in the Crossover Region and Inverse Hall–Petch Region of Nanocrystalline B_4_C

In the Hall–Petch region of B_4_C in Figure 1a, the underlying mechanism is predominantly transgranular fracture mode (nanocrystalline-grained microstructure) [54,60,75]. However, as the grain size approaches the crossover region, intergranular fracture modes (GB-associated microstructures) were also observed [50]. This depicts the possible transition from transgranular fracture mode in the conventional Hall–Petch region to intergranular fracture mode in the inverse Hall–Petch region. To determine mechanical properties in the inverse Hall–Petch region, a resultant shear stress–shear strain curve under finite shear deformation on three varying grain-sized systems of n-B_4_C, GB1—4.84 nm (135,050 atoms), GB2—9.74 nm, and GB3–14.64 nm (3,702,861 atoms), using reactive molecular dynamics (RMD) simulation, were plotted, as in Figure 2a [76,77]. The maximum shear stress obtained was 28.45 GPa for GB1 at 0.5 shear strain, 29.07 GPa for GB2 at 0.33 shear strain, and 29.25 GPa for GB3 at 0.37 shear strain. At large strains, the brittlenesses of GB2 and GB3 were characterized by a smaller plastic deformation range in Figure 2a than GB1 (smaller grain-sized system), which has a larger plastic deformation range indicating its higher ductility. Moreover, through the von Mises shear strain analysis, the dominant deformation mechanism identified in all three models was GB sliding.

Further, local deformation at TJPs to understand GB evolution has been analyzed by calculating shear stress and density variation with respect to shear strain as shown in Figure 2c. Elastic deformation up to the shear strain of 0.2 initially occurred with a constant density. Then, an immediate decrease in shear stress until a 0.225 shear strain followed, before increasing to a strain of 0.275, indicating the annihilation of some pre-distorted icosahedral clusters. A maximum shear stress of 32.23 GPa was obtained at 0.275 shear strain from the region around the TJP, which was higher than that of the complete system. Amorphization caused by stress localization led to the decrease in the density from 0.275 to 0.375 shear strain by relieving shear stress until 11.96 GPa. Then, after the initiation of cavity formation within the amorphous region at 0.4 shear strain, the density decreased dramatically to 0.12 g/m^3^ until a shear strain of 0.475. Through studies on the local deformation at TJPs along with the stress–strain relationship, it was found that an increase in grain size to ~ 15 nm increased the shear strength with distinguished GB sliding.

These intergranular fracture modes typically observed in the inverse Hall–Petch region were verified in n-B_4_C through experimental observations [58,76]. The fracture strength of equiaxed grain sized (~40–150 nm) n-B_4_C micropillars with diameters ranging from 1.5–7 μm is shown in Figure 3a–c [58]. The obtained strength was over 4.5 GPa and reached as high as ~7 GPa using a uniaxial micro-compression test. The plot of fracture strength as a function of sample diameter had no significant effect on the pillar strength indicating an intrinsic property of n-B_4_C, which is not an effect of sample size. The variations in n-B_4_C strength were mainly attributed to the inhomogeneous distribution of nanopores and weak interface phases. Moreover, regardless of diameters, the stress above ~6.5 GPa resulted in a plastic strain of ~0.05–0.08% (in the insert Figure 3a), which was not detected before failure in either single crystals or microcrystalline B_4_C (m-B_4_C). In the scanning electron microscope (SEM) image of uniaxial deformed n-B_4_C micropillars (Figure 3c), the failure proceeded to occur in intergranular mode.

The nanoindentation technique is a widely used method to measure the mechanical properties of even submicron-sized materials and their deformation mechanism. To further understand the mechanical properties, GB sliding, and intergranular fracture mode in n-B_4_C, the nanoindentation technique was used on a focused ion beam (FIB) fabricated cantilever beam specimens shown in Figure 4a–c [76]. Constant loading of 5 mN was applied for 1000 s with a strain rate of ~10^−8^ s^−1^ on the specimen tip (indicated by the red arrow in Figure 4a). The corresponding displacement–time curve in Figure 4c was obtained with linear behavior up to a displacement of ~255 nm, which was followed by a deviation with an average slope of 0.01 nm/s for a long time duration of 1000 s. The sudden small drops in the curve corresponded to the sliding of individual fine grains underneath the indenter indicating that these sliding occurred from high stresses during nanoindentation. Moreover, Figure 4b shows the SEM image of the fractured n-B_4_C cantilever specimen, revealing intergranular fracture mode at a large-sized pore on the surface.

### 2.2. Mechanical Properties in the Crossover Region and Inverse Hall–Petch Region of Nanocrystalline Sic

MD simulation is a powerful computational method to construct nanomaterials ranging a few digits, which are hard to obtain from experimental methods, and to track the occurrence of atomistic mechanisms. Among various empirical potentials developed for classical MD simulations of β-SiC [78,79,80,81,82,83], the potential created by Vashishta [79] and the analytic bond-order potential (ABOP) [78] are the most widely used. These two potentials have shown close agreements with the experimental mechanical properties of β-SiC [84], among which Vashishta potential has shown a smaller deviation from the experimental results compared to ABOP, apart from the shear elastic constant of C_44_ [85]. Moreover, Vashishta potential was able to successfully distinguish between different polymorphs using both MD simulations and density functional theory (DFT) calculations. It is critical to measure these differences in order to accurately describe the microscopic deformation occurring in the system. Thus, MD simulations on n-SiC using Vashishta interatomic potential and LAMMPS package were conducted to measure the mechanical properties in the crossover region [77,79,86]. After the verification of the potential by the reproduction of elastic constants, cohesive energy, melting temperature, and generalized stacking fault energies, various n-SiC systems of grain size ranging from 3.7 nm (325,686 atoms) to 18.6 nm (40,722,382 atoms) with 125 randomly oriented grains were created using the Poisson–Voronoi tessellation method [79,87].

To obtain the shear response of the created n-SiC system, 25 nm length models of cubic n-SiC (1,474,560 atoms) were created with crystal directions of [11$\bar{2}$], [111], [$\bar{1}$10]; [100], [010], [001]; and [110], [$\bar{1}$10], [001] along the x, y, and z directions of the simulation box, respectively, and MD simulations were performed with well-distributed shear strain loading at a constant rate of 10^9^ s^−1^ along xy, xz, and yz planes. Shear stress–strain response and shear strength–yield shear strain (peak stress on the shear stress–strain curve before the drop is defined as shear strength and the resultant shear strain is defined as yield shear strain) response of various grain sizes (3.7, 4.9, 6.2, 7.7, 9.3, 12.4, and 18.6 nm) showed highest shear strength of ~6.5 GPa for grain sizes of 6.2 nm and 7.7 nm. A decrease in grain size increased the shear yield strain ranging from ~0.07 to 0.12 due to the increasing volume of amorphous phases within GBs, as shown in Figure 5b. From two distinctive phases of transgranular crystalline phase and intergranular amorphous phases, consistency with the superplastic tensile behavior of n-SiC [88] was observed by having higher failure strains due to homogenous deformation triggered by controlled plasticity from soft amorphous phases within GBs.

Hall–Petch-like behavior was observed in regions (I) and (II) in Figure 5a, where a sharp increase in shear strength was observed up to ~6.4 GPa in the region (I), plateauing for a grain size of 6.2–9.3 nm in the region (II). Inverse Hall–Petch-like behavior was observed in region (III) (below 6.2 nm of grain size) by a decrease in shear strength with decreasing grain size. Maximum shear strength was obtained at grain size between 6.2 and 7.7 nm due to a combined effect of crystalline volume fraction (in Figure 5b), shear localization process, and GB energies (in Figure 5c). However, in the Hall–Petch region, crystalline volume fraction and total GB energies did not significantly affect the shear strengthening due to the absence of dislocation pile-up strengthening in the simulation, indicating the possibility of other underlying mechanisms.

To understand the mechanism responsible for the Hall–Petch region, shear localization behavior with respect to varying grain size was analyzed in Figure 5d. Through the calculation of the shear localization parameter at the yield shear strain point, two distinct regions were observed, i.e., the high shear localization region (region I) above a grain size of 9.3 nm, where a relatively low strength is observed due to the susceptibility to intergranular nanocracking, fracture, and cavitation, and low shear localization region (region II) below a grain size of 9.3 nm, where shear localization was nearly constant. Assuming that shear deformation is more heterogeneous in the high shear localization region, owing to the increase in volume fraction of crystalline phase with increasing grain size, elastic deformation is initially caused by shear loading energy during shear deformation. The Hall–Petch region has a larger grain size, leading to a higher crystalline volume fraction, which can ultimately lead to elastically strained regions in GBs along with lower stress relaxation and restriction of shear plastic flow due to high constraint. Therefore, deformation after critical shear strain energy leads to intergranular fracture subsequent to cavitation, shear localization, bond breaking, nanocrack formation, and GB sliding due to the cohesion strength of GBs [76]. On the other hand, a decrease in grain size leads to an increase in density of GBs, leading to lower delocalized shear stress applied in the structure that results in homogenous shear deformation, resulting in increased shear strength, along with minimized stress concentration at GBs, and delocalized shear flow. Thus, this also indicates that the shear response in the inverse Hall–Petch region (region III in Figure 5a) is weaker and softer due to the increased volume fraction of the disordered phase in GBs. Moreover, the shear strains were localized mainly along GBs, indicating that failure and deformation in n-SiC in all the regions (Hall–Petch and inverse Hall–Petch region) were mainly GB-mediated.

Through von Mises local atomic shear strain analysis, deformation type (heterogeneous or homogenous) and shear localization were further validated for a grain size of 3.7 nm, representative of finer grain, and grain size of 12.4 nm, representative of coarse grains [89]. For the grain size of 12.4 nm, high stress was concentrated at GBs and TJPs, leading to the relatively low strength by premature failure, and for the grain size of 3.7 nm, shear stress was homogeneously distributed throughout the sample and minimally intensified at GBs leading to the higher critical shear strain by delayed failure and homogenous deformation. Thus, in correspondence with the analysis in Figure 5d, the failure and deformation of n-SiC were led by GB-mediated mechanisms due to shear strain energy release. In fact, in another study [90], MD simulation verified the experimental results and demonstrated that the formation of amorphous bands in β-SiC is due to the coalescence of the pile-up of small stacking faults and the growth of new stacking faults from old ones when the longitudinal shock was induced. Therefore, these results suggest that the formation of amorphous bands is prone to failure, as they can create new interfaces, providing sites for crack nucleation and growth.

Further analysis of the crossover region was conducted through the development of an appropriate composite model (CM) of n-SiC by varying grain shapes (cubic, spherical and tetrakaidekahedral) with varying GB thicknesses of 0.5–1 nm to describe shear strength using Hill’s like model [91]. Grain shapes of cubic and spherical showed closest fitting to the crystalline volume fraction curve with GB thickness of 0.75 nm. Upon the variation of GB thickness in the model of combined cubic and spherically shaped grains, a critical grain size of ~6.3 nm at maximum shear strength of around 5.8 GPa was obtained using theoretical CM calculations. This correctly matched the critical grain size of 6.2–7.7 nm in MD results where the transition occurred. Moreover, in contrast to the observed independence of GB thickness from the grain size in MD, an increase in GB thickness with an increase in grain size, and the diminished importance of GB thickness on the shear strength in the Hall–Petch region (typically above grain size of 100 nm, GB thickness has no impact on shear strength), were identified through the CM calculations.

To shed light on the atomistic mechanism responsible for the mechanical response of n-SiC occurring at the crossover region, the load–displacement curve in Figure 6a was obtained using the nanoindentation technique on MD simulated n-SiC [92]. Four different regimes were identified with the transition depth (h_CR_) at 14.5 Å. Due to amorphous GBs that were coupled together, a collective inter- and transgranular response was observed in regimes 1 and 2, where elastic regime 1 slowly showed plastic yield as the depth approached h_CR_. This collective response involved both couplings of grains outside the indentation region and the formation of mesoscopic shear planes. Above the h_CR_, in regime 3, amorphous GBs started to plastically yield with the decoupling of grains containing nanopores. Crystalline phases of grains started only to yield after regime 4. The observed phenomenon was particular to nanocrystalline materials, where the nucleation of dislocations occurs with yielding. Further understanding of the transition between the collective response and pure intergranular response was observed from average displacement to the center of mass of grains versus indentation depth curve in Figure 6b. With the coupling of the grains until the h_CR_ in regime 1, the coupling rate decreases in regime 2 where it reaches a minimum in regime 3. In regime 4, grains are decoupled, and oscillation can be observed corresponding to the small drops in stress in regime 4 in Figure 6a. These drops are related to grain rotations, grain sliding, dislocation formation in the grains, and coupling of grains underneath the indenter. Through the demonstration of deformation localization and crossover response in the grain rotation, the percentage of the disordered atom-displacement curve was calculated in Figure 6c. Regimes 1 and 2 were characterized by crystallization-dominated deformation, whereas regimes 3 and 4 were characterized by disordering-dominated deformation, indicating the transition from transgranular (crystalline) and intergranular (disordered) responses.

Indeed, these intergranular responses with non-elastic deformation were observed in a uniaxial micro-compression test on ~10–100 nm grain-sized n-SiC [69]. A fracture strength of over 8 GPa was obtained on varying diameters (1–13.5 μm) of n-SiC micropillars in Figure 7a–c, where maximum strength reached as high as ~11 GPa. The obtained values were significantly higher than those of microcrystalline β-SiC (m-SiC), which has a strength of about 2–4 GPa [93,94]. Thus, in addition to outlining the importance of nanocrystalline, homogenous, and dense ceramics, understanding the underlying deformation mechanism in the enhancement of its mechanical properties at the atomic level will be discussed in the following section.

## 3. Deformation Behavior in Nanocrystalline B_4_C and β-SiC

### 3.1. Nanocrystalline B_4_C

The spark plasma sintering (SPS) method and isostatic pressing, such as cold- and hot-isostatic pressing method (CIP and HIP), can yield dense B_4_C and SiC compared to the conventional hot-pressing method [8,48,95,96]. These methods are efficient in densifying B_4_C and SiC by eliminating the porosity present in ceramics, which degrades their mechanical properties by acting as a crack initiator [53]. Based on this understanding, the HIP method was used to fabricate n-B_4_C and n-SiC, using the same sintering parameters of a relatively low sintering temperature of 1600 °C and pressure of 980 MPa to avoid grain coarsening [58,69]. The obtained n-B_4_C sample reached 92.8% of theoretical density with a mean grain diameter of ~32 ± 7 nm, and homogeneously distributed irregular-shaped nanopores ranging from 20 to 70 nm were located at GBs and TJPs in Figure 8a [58]. GBs and the internal surface of nanopores were confirmed by high-resolution TEM (HRTEM) images and fast Fourier transformation (FFT) pattern, shown in Figure 8b, which were surrounded by a thin layer of amorphous carbon interface. Moreover, Figure 8c confirms the clean GBs without interface phases. The microstructural changes after indentation were studied through TEM analysis on FIB fabricated cross-sectional slice of residual indent, as shown in Figure 8e–f [58]. Reductions in the size and population of nanopores were observed in the deformed region of the specimen at the maximum load of 1000 mN by densification through indentation loading force when compared to the neighboring undeformed region. The inset of selected area electron diffraction (SAED) patterns of the undeformed and deformed region (Figure 8b,c, respectively) shows two similar patterns, indicating that there is no crystal-to-crystal phase transition that happened during deformation. Moreover, an intrinsic toughening mechanism was observed through no observable cracks within the deformed region. Pure intergranular fracture mode in n-B_4_C was identified through the winding crack deflection in the residual indentation corner. These observations demonstrate GB sliding by nanoporosity elimination and disordered boundary phases, and the observed phenomenon was in contrast to m-B_4_C, where failure proceeds by transgranular amorphization and cleavage during fracture [19,97].

Interestingly, varying the composition in the varying thickness of B_4_C amorphous interface phases within the deformed regions was observed by bright-field scanning TEM (BF-STEM) with confirmation through separate electron energy loss spectroscopy (EELS) spectrum in Figure 9a–d. The thinner amorphous interface phase (Figure 9c,d) had a chemical composition similar to B_4_C, and the thicker amorphous interface phase (Figure 9a,b) had a more significant content of carbon than crystalline B_4_C. This indicated that the formation of deformation-induced amorphous B_4_C occurs during GB sliding, along with the elimination of nanopores at clean GBs. This helps in enhancing the plasticity of brittle B_4_C by providing lubrication at GBs to activate GB sliding. Thus, the enhanced toughness of n-B_4_C is related to GB sliding along thin amorphous carbon layers and nanopore elimination, where crack initiation is inhibited to activate local plastic deformation [92,98].

The detailed deformation process in n-B_4_C [76] at atomic level is shown in Figure 10a–c by atomic strain analysis using von Mises shear strain [89]. Elastic deformation is observed in the initial deformation strains where the magnitude of shear stress increases linearly at a constant rate up to 0.25, 0.275, and 0.3 for GB1 (4.84 nm), GB2 (9.74 nm), and GB3 (14.64 nm), respectively, without significant local deformation (Figure 10(a1–c1)). Moreover, dominant deformation was identified in GB regions in all three models, with most slipped atoms located within the GB region. After reaching a critical shear strain, amorphous bands were observed along GBs, which relieved the shear stress (Figure 10(b2,c2)). Crack initiation and cavitation were initiated within these amorphous bands, and shear stress was further relaxed by crack opening (Figure 10(b3,c3)). These crack openings in three GB models indicated that the major fracture mode in the n-B_4_C was an intergranular fracture.

### 3.2. Nanocrystalline β-SiC

Similar to n-B_4_C, HIP-fabricated bulk n-SiC had a theoretical density of 97%, and the nanograin size ranged from ~10–100 nm, which were randomly distributed, showing nearly fully dense n-SiC [69]. The measured mean and standard deviation of grain sizes were 35 ± 15 nm, with ~0.68–8 nm wide graphitic carbon, which were uniformly encapsulated at GBs and TJPs, as shown in Figure 11a. Moreover, similar to what was observed in [99], high densities of stacking faults and nanotwins were observed. However, no dislocations were observed in these samples. It is, therefore, suggested from these undeformed microstructural observations that grain growth is restricted by low sintering temperature and the presence of a graphitic carbon nanolayer phase at GBs and TJPs.

To further understand the underlying deformation failure mechanism in n-SiC, the cross-sectional specimen of residual indent fabricated by FIB was analyzed using TEM, as shown in Figure 11d–f [18,69,100]. The indented regions in the specimen at a maximum load of 1000 mN showed a diffusive interface in Figure 11d,e, which was in contrast to the undeformed region that had a sharp interface in Figure 11a–c. The grain sizes in the deformed region were less than 20 nm in clusters (marked by the red circles in Figure 11d) that were free of carbon interface phases with small pores allowing grain sliding. Intrinsic toughening was also observed through shear sliding due to high stresses, as no observable cracks within the indent region were found [23,58]. Moreover, it can be seen from the BF-STEM images in Figure 11e–f that shear deformation accompanied by GB sliding induced the transformation of graphitic carbon to highly disordered carbon at GBs and TJPs. Interface along <$\bar{1}10$> direction in the deformed region was analyzed by STEM with FFT (in Figure 11f), which showed the emission of dislocation along with the ($11\bar{1}$) glide plane. This observation indicated that during shear sliding deformation, high local shear stresses are produced. Further load transfer and mobility of grain entities during GB sliding were confirmed through the production of dislocation in the proximity of the GB interface. Grain sliding during shear deformation occurred, leading to a redistribution of internal stresses by dislocation-induced plasticity. Thus, crack initiation and failure are delayed through the accommodation of local plastic deformation by substantial grain sliding and carbon interface phases at high stress.

As suggested in [99], observed intergranular fracture arises from the initiation mechanism by a build-up of stress concentration at stacking faults and twinning, and at GB with secondary phases. Moreover, the obtained observation is in contrast with the m-SiC [23,101,102,103], where crack propagation is usually accompanied by transgranular fracture mode with its characteristic brittle cleavage. The underlying mechanism to understand enhancement in strength and plasticity was investigated using TEM on the deformed region after uniaxial deformation revealing high strain in the amorphous carbon interface phases compared to before deformation. Structural changes in proximity or at GBs and TJPs in the form of high strain in the interface were identified in the deformed region from the BF-STEM image. Moreover, dislocation emission along {111} plane indicated that before crack initiation, shear stresses are resolved by yielding plastically during high-stressed deformation.

Deformation on laminated n-SiC and graphite (carbon) structures under finite shear deformation along the (111) <$\bar{2}11$> plane was carried out using DFT-based MD simulations to understand the deformation process at the atomic level [69]. Simulated uniaxial compression at 10% and 15% based on the indentation experimental results were modeled perpendicular to the slip plane at a constant shear rate, 0.02 ps^−1^. The shear stress–strain curve of shear deformation at 10% compression and 19.07 GPa compressive stress along the z-axis resulted in an increase in shear stress from zero to the maximum value of 22.58 GPa at 0.67 shear strain. No significant local deformation was observed during the whole shear process with elastic deformation in the overall system before 0.2 shear strain. This was supported by the observed small drops in the curve, which occurred from the ease of carbon interface phase sliding under shearing, preventing plastic deformation of the SiC structure and crack initiation due to the relaxation of the system stress by transfer of loads in n-SiC to laminar graphite.

At 15% compression (37.05 GPa along the z-axis) in Figure 12a, an increase in shear stress up to 14.87 GPa with an increase in shear strain to 0.46 was observed with varying rates of increase due to load transfer and interface sliding. Out-of-plane deformation by a slight shift along the z-direction occurred from laminar carbon sliding and a slight shift in carbon atoms of different stacking carbon layers and interfacial n-SiC, as shown in Figure 12b,c. This indicates the presence of strong van der Waals interaction between the adjacent carbon atoms. Phase transformation from the graphite to diamond phase was indicated by the sharp decrease in shear stress at 0.49 shear strain due to the formation of new carbon bonding from neighboring graphite layers and n-SiC layers with the shifted carbon atoms, as shown in Figure 12d. This provided an easy method of transformation, which was observed at a lower transition shear stress of 14.87 GPa under uniaxial stress of 44.02 GPa than previously reported DFT calculations [104]. Subsequently, dislocations were observed in n-SiC (Figure 12e) by an increase in shear stress and relaxation, which occurred with a gradual increase in shear strain to 0.62 before the bond breakage between Si and C atoms, which occurred at 0.70 shear strain. The observed phase transition in n-SiC was different from the monocrystalline β-SiC substrate reported by Zhao et al., where the MD simulation of nanoindentation on monocrystalline β-SiC (111) underwent a phase transition from cubic diamond to amorphous phase [85]. In addition, elastic–plastic transitions were observed with ‘pop-in’ during loading at the displacement of 1.57 nm with reversible amorphization-induced quasi-elastic deformation.

## 4. Summary and Outlook

Superhard lightweight ceramics such as B_4_C, β-SiC, etc., have exceptional mechanical properties. However, due to their innate nature, i.e., rigid covalent and ionic bonding, they undergo brittle fractures with low toughness and strength. The mechanical properties of these ceramics showed strong dependence on composition variation, anisotropy, microstructural features, and sintering conditions. Pressurized sintering such as SPS and the isostatic pressing method show enhanced mechanical properties than the conventional hot-pressing method or pressureless-sintering methods due to the restriction of grain growth and high density of over 95% along with nanoporosity. Moreover, the presence of highly disordered carbon interface phases at GBs and TJPs with nanopores in nanocrystalline ceramics has been observed to contribute to significant plastic deformation, promoting GB sliding. GB sliding is easily induced by salient microfeatures of nanosized grain and nanoporosity, enabling amorphization at GBs and TJPs and intergranular fracture mode, leading to an increase in strength, plasticity, and fracture toughness. The observed mechanism is in contrast to that in microcrystalline ceramics, where porosity and interface phases can lead to catastrophic failure by the transgranular fracture mode. Consistent with the experimental results in nanocrystalline ceramics, theoretical studies performed using ReaxFF-based and classical MD simulations showed detailed GB deformation at the atomistic level, confirming the amorphous band formation by GB sliding. Thus, evidence of the enhanced mechanical properties in brittle ceramics through controlled microfeatures and the underlying mechanism behind the inverse Hall–Petch region have been highlighted in this review based on the current research progress.

No direct evidence of enhanced mechanical properties through GB sliding and transition between the Hall–Petch and inverse Hall–Petch relationships in other superhard lightweight ceramics have yet been identified. Moreover, HPHT-sintering conditions have proven to be effective in synthesizing superhard cBN and β-SiC, which still need to be explored for B_4_C. Nevertheless, from the increased mechanical properties observed in HPHT-sintered cBN and β-SiC, B_4_C sintered under these conditions can have its hardness increased, possibly surpassing the superhard threshold by reduced grain growth and atomic diffusion, which are critical limitations for superhard B_4_C. Moreover, the observed formation of an amorphous interface by GB sliding in B_4_C and SiC in this review suggests that these mechanisms can be potentially extendable to other ceramics such as Al_2_O_3_, MgAl_2_O_4_, and Si_3_N_4_.

## Figures and Tables

**Figure 1 nanomaterials-12-03228-f001:**
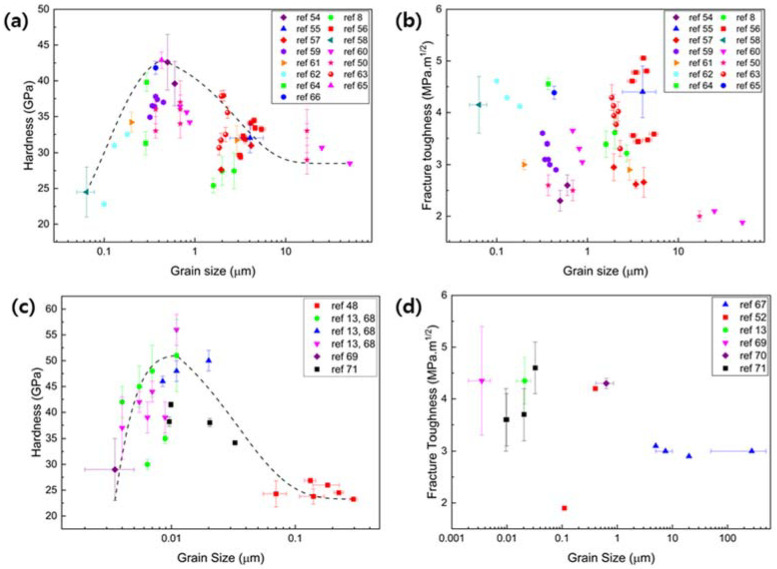
Grain size dependence curve of, (**a**) hardness of B_4_C, (**b**) fracture toughness of B_4_C [8,50,54,55,56,57,58,59,60,61,62,63,64,65,66], (**c**) hardness of β-SiC, and (**d**) fracture toughness of β-SiC [13,48,52,67,68,69,70,71].

**Figure 2 nanomaterials-12-03228-f002:**
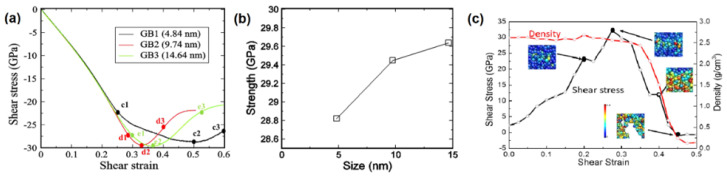
(**a**) Three n-B_4_C GB models under finite shear deformation at 0.1 ps^−1^ shear rate. (**b**) The shear strength–size curve of three n-B_4_C GB models. (**c**) Shear stress and density–strain curve at TJPs. Reproduced with permission from [76]. Copyright 2022 APS.

**Figure 3 nanomaterials-12-03228-f003:**
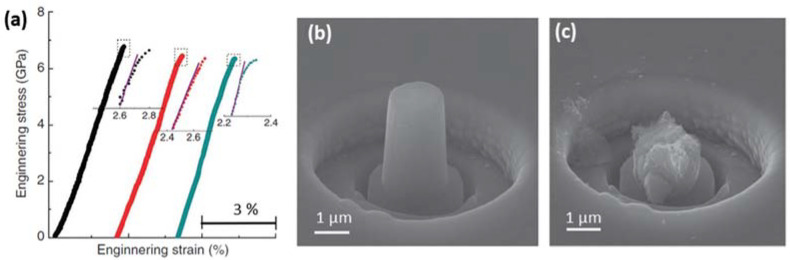
(**a**) Engineering stress–strain curve of n-B_4_C micropillars obtained by uniaxial micro-compression test (Colors of the curves correspond to the micropillar diameter of 1.6 μm for black, 2 μm for red, and 7 μm for green). (**b**) SEM image showing fabricated n-B_4_C micropillar with a diameter of 1.6 μm. (**c**) SEM image showing fractured n-B_4_C micropillar by intergranular mode. Reproduced with permission from [58]. Copyright 2022 Nature Communications.

**Figure 4 nanomaterials-12-03228-f004:**
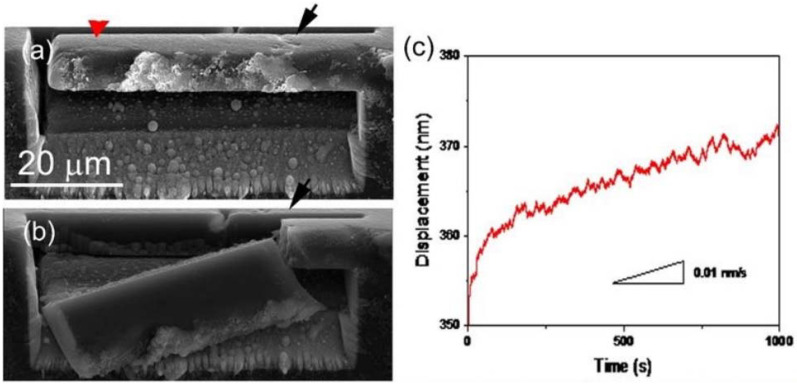
(**a**) SEM micrograph of representative n-B_4_C cantilever showing some nanopores (indicated by the black arrow) on the surface near the pivot. (**b**) SEM micrograph of fractured n-B_4_C cantilever upon indentation loading force of 6.145 mN at red arrow in Figure 4a. (**c**) The displacement–time curve of the n-B_4_C cantilever at constant loading of 5 mN shows permanent deformation with a slope of 0.01 nm/s. Reproduced with permission from [76]. Copyright 2022 APS.

**Figure 5 nanomaterials-12-03228-f005:**
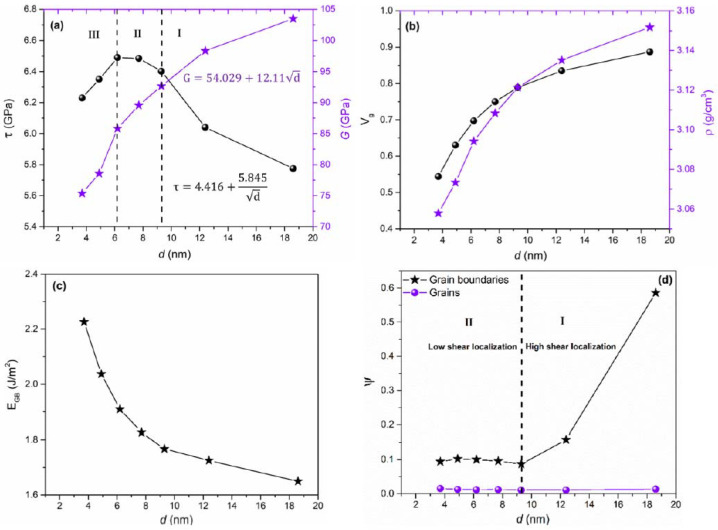
(**a**) Shear strength (τ) and modulus (G) (given in black circle and purple star respectively), (**b**) crystalline volume fraction (V_g_) and density (ρ) (given in black circle and purple star respectively), (**c**) total grain boundary energies (E_GB_), and (**d**) shear localization parameters (ψ) as a function of grain size. Reproduced with permission from [86]. Copyright 2022 APS.

**Figure 6 nanomaterials-12-03228-f006:**
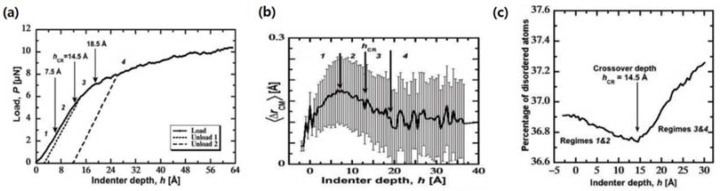
(**a**) Load–displacement curve of n-SiC. (**b**) Average displacement with standard deviation to the center of mass of grains with respect to indentation depth curve. (**c**) Percentage of the disordered atoms–displacement curve. Reproduced with permission [92]. Copyright 2022 Science.

**Figure 7 nanomaterials-12-03228-f007:**
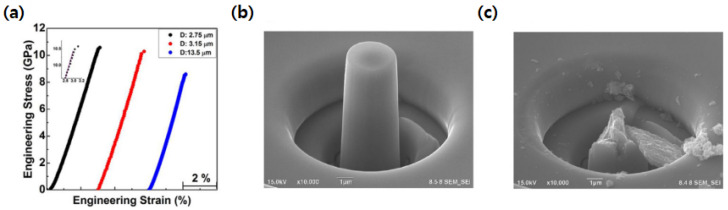
(**a**) Engineering stress–strain curve of n-SiC micropillars obtained by compression test. (**b**) Scanning electron micrograph showing fabricated n-SiC micropillar with a diameter of 1.6 μm. (**c**) Scanning electron micrograph showing fractured n-SiC micropillar by intergranular mode. Reproduced with permission from [69]. Copyright 2022 Elsevier.

**Figure 8 nanomaterials-12-03228-f008:**
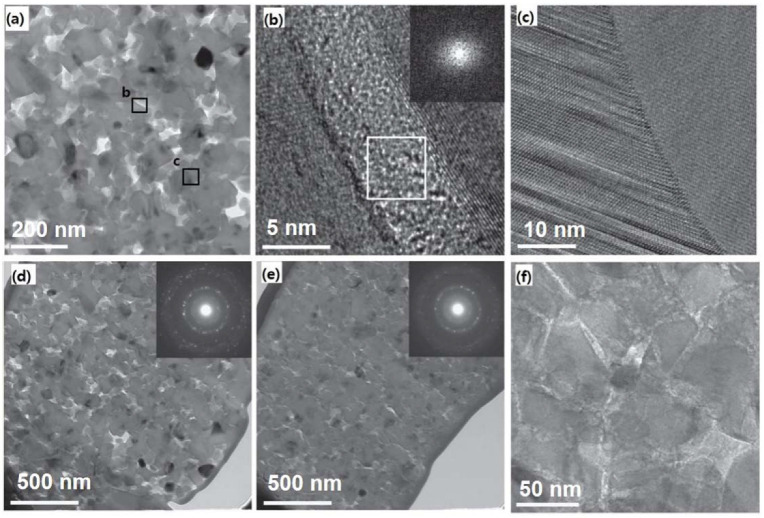
(**a**) TEM image of n-B_4_C showing homogeneously distributed nanopores (Figure 8b,c correspond to the respective boxes indicated in (**a**)). (**b**) HRTEM image of GB showing the presence of amorphous carbon interface phase between two grains and the corresponding FFT pattern (given in the inset). (**c**) HRTEM image of clean GB showing absence of interface phase. (**d**) BF-TEM micrograph showing uniformly distributed pores in n-B_4_C under the undeformed region and the corresponding SAED pattern (given in the inset). (**e**) BF-TEM image showing densification of n-B_4_C under deformed region and the corresponding SAED pattern (given in the inset). (**f**) TEM micrograph showing well-distributed encapsulation of interface carbon around n-B_4_C grains. Reproduced with permission from [58]. Copyright 2022 Nature Communications.

**Figure 9 nanomaterials-12-03228-f009:**
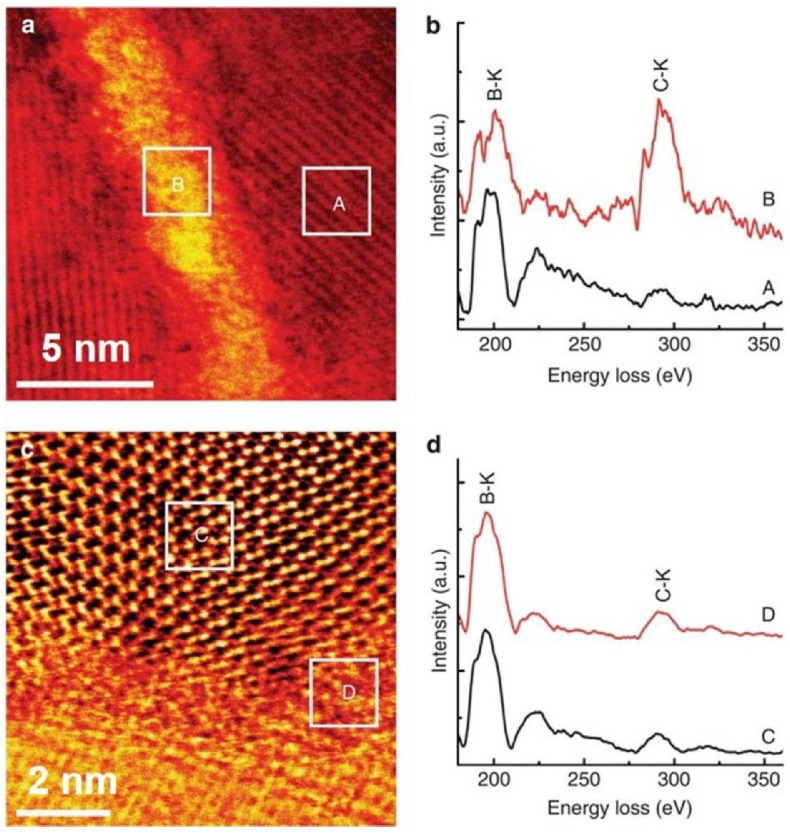
(**a**) BF-STEM image showing broad GB between two grains. (**b**) EELS spectrums corresponding to boxes in Figure 9a. (**c**) BF-STEM image showing narrow GB between two grains. (**d**) EELS spectrums corresponding to boxes in Figure 9c. Reproduced with permission from [58]. Copyright 2022 Nature Communications.

**Figure 10 nanomaterials-12-03228-f010:**
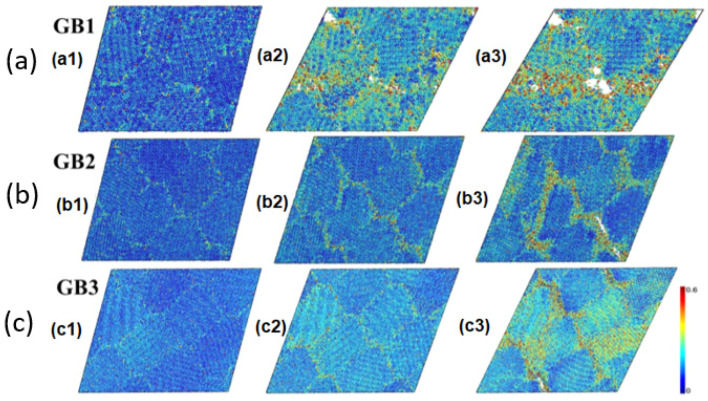
(**a**–**c**) Snapshots of three n-B_4_C GB models at different time intervals, showing (**a1**–**c1**) initiation of plastic deformation, (**a2**–**c2**) termination of plastic deformation, and (**a3**–**c3**) formation of cavities. Strain from 0.0 to 0.6 is represented by the colored bar code. Reproduced with permission from [76]. Copyright 2022 APS.

**Figure 11 nanomaterials-12-03228-f011:**
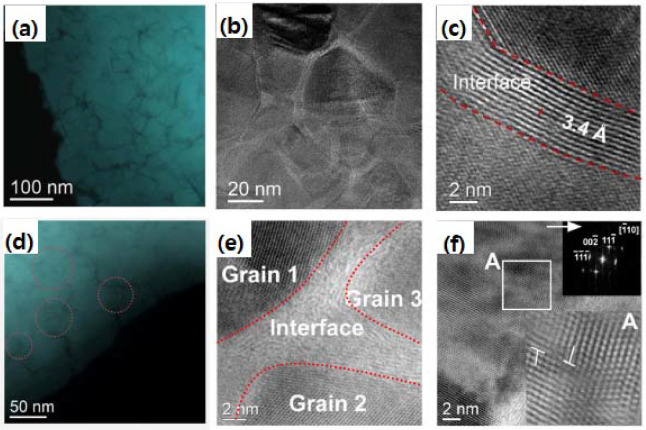
(**a**) Dark-field STEM and (**b**) magnified TEM image, showing n-SiC grains surrounded by evenly distributed carbon at GB. (**c**) HRTEM image of pristine n-SiC, showing graphitic lattice of carbon interface at GB. (**d**) Dark-field STEM image after deformation of n-SiC by indentation. (**e**) BF-STEM micrograph of n-SiC after deformation, showing highly disordered carbon interface at GB. (**f**) BF-STEM micrograph on [110] direction of n-SiC, showing dislocation along {111} plane and the corresponding FFT pattern (given in the inset). Reproduced with permission from [69]. Copyright 2022 Elsevier.

**Figure 12 nanomaterials-12-03228-f012:**
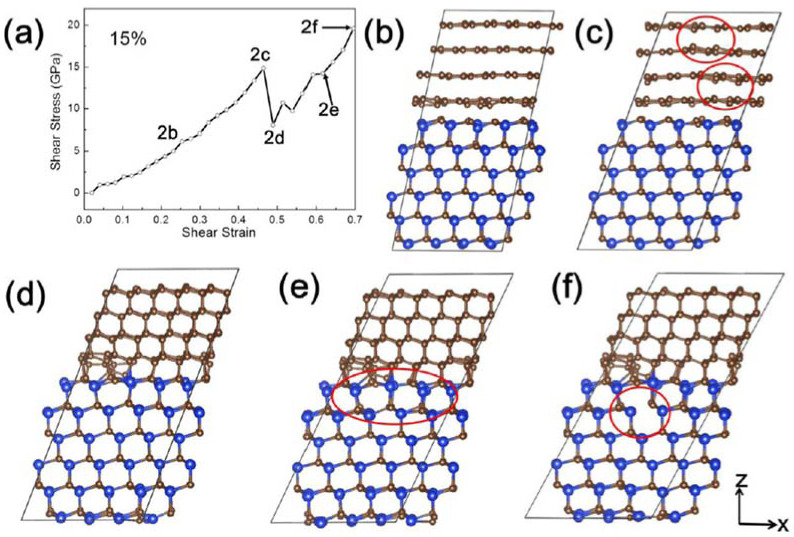
(**a**) Shear stress–strain curve of SiC and carbon integrated system at 15% pre-compressed rate along the z-axis. (**b**–**f**) Snapshots of structures of the integrated system corresponding to the indications in Figure 12a. (**b**,**c**) A shift of carbon atoms to the neighboring carbon layer. (**d**,**e**) Formation of diamond structured phase by new bond formation of SiC and carbon, and dislocation in SiC. (**f**) Breakage of bonding in n-SiC at the shear strain of 0.696. Reproduced with permission from [69]. Copyright 2022 Elsevier.

## Data Availability

Not applicable.

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
