# Peer review of "Mechanical Properties and Deformation Behavior of Superhard Lightweight Nanocrystalline Ceramics"

_nanomaterials, 2022, doi:10.3390/nano12183228_

Round 1

Reviewer 1 Report

Jeong et al. provide a review of recent work on nanocrystalline B4C and b-SiC as related to deformation in the transition region from the Hall-Petch and inverse Hall-Petch regimes. They do so with a mix of mechanical studies, transmission electron microscopy and modeling. Generally review papers incorporate a broad spectrum of relevant work from other authors. However, this review largely focuses on four papers from the authors of this manuscript, specifically references 45, 57, 61, 72. Of the 12 figures in the manuscript, 10 are from the authors’ own papers. I find this unacceptable as a legitimate review paper. 

In addition to this general comment, there area specific questions and comments: 

Pages 3 and 4, Figure 1 (b) and (d): With what confidence is the toughness increasing with decreasing grain size for either material? The data should be evaluated statistically, and error bars should be included on slopes, if slopes are warranted. Given that these data are from many different experiments and laboratories, true comparisons are difficult. Therefore, stating that toughness has increased as the grain size has decreased is somewhat suspect. 

Page 4, line 139, 151: Why is MgAl2O4 included in this discussion? The authors should say upfront that they will be comparing B4C and b-SiC to MgAl2O4 and why.

Page 4, line 167-68: Why should the shear region be the smallest for mid-size grain?

Page 4, Figure 2(c): How does this figure show the impact of the TJP? How was this analyzed?

Page 6, Figure 3(a): A better description of this figure is needed. What do the colors represent?

Page 7/8, Figure 5(b): How is crystallinity established for n-SiC in the model? Is the modeling of crystallinity ever verified with experiments?

In the summary and outlook, there is little discussion of unanswered questions.

Author Response

Manuscript ID: nanomaterials-1894930

 Response to Referee’s-1 comments:

Jeong et al. provide a review of recent work on nanocrystalline B4C and b-SiC as related to deformation in the transition region from the Hall-Petch and inverse Hall-Petch regimes. They do so with a mix of mechanical studies, transmission electron microscopy and modeling. Generally review papers incorporate a broad spectrum of relevant work from other authors. However, this review largely focuses on four papers from the authors of this manuscript, specifically references 45, 57, 61, 72. Of the 12 figures in the manuscript, 10 are from the authors’ own papers. I find this unacceptable as a legitimate review paper. 

Response: We would like thank reviewer for his/her comments and suggestions for our review manuscript. Authors are aware that the review papers need to incorporate a broader spectrum of relevant work in the research field. However, to meet the scope of the special issue (Superhard materials with nanostructures) in Nanomaterials, we have selected superhard ceramics which were investigated deeply with regard to their deformation mechanisms and their transition from Hall-Petch to inverse Hall-Petch relationship. Boron carbide and silicon carbide were considered to be superhard ceramics that have enough supporting evidence and understanding to meet this criterion. In addition, we have introduced briefly other researchers’ work on superhard materials in Sections 1 and 2 in the revised review manuscript.

In addition to this general comment, there area specific questions and comments: 

 Pages 3 and 4, Figure 1 (b) and (d): With what confidence is the toughness increasing with decreasing grain size for either material? The data should be evaluated statistically, and error bars should be included on slopes, if slopes are warranted. Given that these data are from many different experiments and laboratories, true comparisons are difficult. Therefore, stating that toughness has increased as the grain size has decreased is somewhat suspect. 

Response: In previous version, slopes were added for toughness graphs and the authors agree with reviewer that the addition of slopes and the general trend-lines in the corresponding graphs can provide misunderstandings to the readers. Thus, we have removed the slopes in Figures 1 (b) and (d). Moreover, the discussion has been focused more clearly on the variation of hardness with grain size, as interpreted from Figure 1. We have explained in the text that hardness depends on the grain size while fracture toughness shows independence

Page 4, line 139, 151: Why is MgAl2O4 included in this discussion? The authors should say upfront that they will be comparing B4C and b-SiC to MgAl2O4 and why.

Response: We agree with reviewer’s comment that the inclusion of MgAl2O4 in Page 4, line 139-151 is not suitable for the direct comparison for the Hall-Petch and inverse Hall-Petch relationship because MgAl2O4 is not a superhard ceramic like SiC or B4C, thus, the authors have removed previous lines 139-151.

Page 4, line 167-68: Why should the shear region be the smallest for mid-size grain?

Response: Larger-grain-sized GB 2 and 3 models have a smaller plastic deformation range due to less fractured icosahedra at grain boundaries. This was consistent with the study on the grain boundary by quantum mechanics calculations in [1] where the lower grain boundary energy model is more brittle than the higher grain boundary energy model.

Page 4, Figure 2(c): How does this figure show the impact of the TJP? How was this analyzed?

Response: We thank the reviewer for pointing this out. We have corrected the word ‘impact’ to ‘local deformation’ to clarify the text because the Figure. 2 (c) describes local deformation at the TJPs in GB 2 model.

Page 6, Figure 3(a): A better description of this figure is needed. What do the colors represent?

Response: Colors are represented to distinguish the stress-strain curves of different sizes of micropillars. The authors have made an explanation in revised manuscript, lines 229-230.

Page 7/8, Figure 5(b): How is crystallinity established for n-SiC in the model? Is the modeling of crystallinity ever verified with experiments?

Response: The polycrystalline structure of n-SiC was established using the Poisson-Voronoi tessellation method using Vashishta interatomic potential based on the n-SiC models from I. Szlufarska [2]. Moreover, n-SiC models created by I. Szlufarska were verified with experimental results from [3].

In the summary and outlook, there is little discussion of unanswered questions.

Response: The authors have added further information in the outlook section for readers to discuss more on unanswered questions (In revised manuscript, lines 612-619).

References

  1. Yang, X., et al., Shear-Induced Brittle Failure along Grain Boundaries in Boron Carbide. ACS Applied Materials & Interfaces, 2018. 10(5): p. 5072-5080.
  2. Mo, Y. and I. Szlufarska, Simultaneous enhancement of toughness, ductility, and strength of nanocrystalline ceramics at high strain-rates. Applied Physics Letters, 2007. 90(18): p. 181926.
  3. Chen, D., et al., Role of the grain-boundary phase on the elevated-temperature strength, toughness, fatigue and creep resistance of silicon carbide sintered with Al, B and C. Acta Materialia, 2000. 48(18): p. 4599-4608.

Reviewer 2 Report

The manuscript "Mechanical properties and deformation behavior of superhard light-weight nanocrystalline ceramics" (authors: Byeongyun Jeong, Simanta Lahkar, Qi An, Kolan Madhav Reddy) corresponds to the scope of "Nanomaterials". A study of two methods for obtaining carbon coatings, which are compared with each other, is shown. The manuscript has a lot of comments, so it requires minor revisions.

1. In Introduction. When describing the properties of ceramics, in particular superhard, strong, etc., it is worth providing characteristic values ​​of the indicated quantities. Superhard materials typically have a hardness of more than 40 GPa. What is the value in this case?

2. What phase or type of crystal lattice did the authors mean by "…microcrystalline B4C (m-B4C)…"? Does the index "m" mean "microcrystalline" or something else, e.g."monoclinic syngony"? An explanation is required here. It is also worth noting that when describing silicon carbide, a specific phase (β-SiC) was indicated. It is worth using one type of description of crystalline materials, at least within one paragraph.

3. In the review, the Hall-Petch relationship can be presented as a formula, which will allow readers to better understand the features of this pattern.

4. Lines 91-92. Here is a mixed designation of phases again, which is related to the grain size and to the type of crystal lattice. See point 2. It is worth moving this part with the designations of this type to section 2.

5. Based on fig. 1(a,b) it can be concluded that B4C has a very limited region of existence of the superhard state (more than 40 GPa). Is it then worth saying that this ceramic is superhard?

6. Fig. 1(d). There are significant deviations of the points from the linear trend. What is the statistical value of this approximation?

7. Letters (a, b, c) in figures 3 and 7 must be made the same.

8. p.16, line 525. The value "... 0.02 ps-1" is better to indicate with an exponent for easier understanding.

Author Response

Manuscript ID: nanomaterials-1894930

 Response to Referee’s-2 comments:

  1. In Introduction. When describing the properties of ceramics, in particular superhard, strong, etc., it is worth providing characteristic values ​​of the indicated quantities. Superhard materials typically have a hardness of more than 40 GPa. What is the value in this case?

Response: Thank you for your suggestions. We have added a new paragraph in lines 26-42 introducing superhard ceramics with the characteristic values of B4C and SiC along with representative superhard ceramic, cubic boron nitride.

  1. What phase or type of crystal lattice did the authors mean by "…microcrystalline B4C (m-B4C)…"? Does the index "m" mean "microcrystalline" or something else, e.g."monoclinic syngony"? An explanation is required here. It is also worth noting that when describing silicon carbide, a specific phase (β-SiC) was indicated. It is worth using one type of description of crystalline materials, at least within one paragraph.

Response: “m-B4C” indicates the microcrystalline B4C in our paper. According to the reviewer’s suggestion, we have edited to meet the needs of using one type of description of crystalline materials in the manuscript.

  1. In the review, the Hall-Petch relationship can be presented as a formula, which will allow readers to better understand the features of this pattern.

Response: We thank the reviewer for the suggestion. We have added the equation expressing the general Hall-Petch relationship in lines 78-80.

  1. Lines 91-92. Here is a mixed designation of phases again, which is related to the grain size and to the type of crystal lattice. See point 2. It is worth moving this part with the designations of this type to section 2.

Response: We have removed lines 91-92 (in previous version), which show the anisotropy of hexagonal SiC which is different crystal lattice than what is described mainly in the manuscript. Instead, the authors have added new lines (In the revised manuscript corresponding to lines 121-122) showing anisotropy in cubic (b-SiC) to be in consistency with the crystal lattice of SiC discussed throughout the manuscript. Moreover, we have edited previous lines 91-92 (In the revised manuscript corresponds to lines 109-111) expressing nanocrystalline SiC rather than a mixed designation of phases i.e., grain size and crystal lattice.

  1. Based on fig. 1(a,b) it can be concluded that B4C has a very limited region of existence of the superhard state (more than 40 GPa). Is it then worth saying that this ceramic is superhard?

Response: Yes, we agree with reviewer’s comments that boron carbide has a limited region of existence of the superhard state (more than 40 GPa). Commercial silicon carbide [1-3] has relatively lower hardness (20-27 GPa) compare to commercial boron carbide (25-30 GPa) [4]. The commercial boron carbide consists of large grain size in the range from 2 mm to 25 mm and secondary phases. The literature data in Figure. 1. (a), which were chosen without any sintering additives, suffer lower hardness (less than 40 GPa) due to the presence of porosity and grain growth. Moreover, there is yet no existing literature on HPHT (pressure >5 GPa) synthesized boron carbide. Recently it was demonstrated that silicon carbide can reach high hardness of 41.5 GPa when it was fabricated under high pressure and high temperature (HPHT) sintering conditions. In addition, both silicon carbide [5] and boron carbide [6] in their thin film form have been reported to have hardness over 50 GPa. Moreover, B4.3C in its single crystal crystals showed hardness over 45 GPa exceeding the superhard threshold (< 40 GPa) [7]. Thus, we believe that boron carbide could be superhard ceramic, which is third hardest material after diamond and cubic boron nitride.

  1. Fig. 1(d). There are significant deviations of the points from the linear trend. What is the statistical value of this approximation?

Response: Linear trend lines in Figure. 1 (b) and (d) were added for reference to readers for them to interpret the data more easily. Moreover, there is a significant deviation in Figure. 1 (d) compared to Figure. 1 (b) due to the limited experimental data available for silicon carbide. However, according to the comment and suggestion from Reviewer 1, we have removed the slope because it can lead to misunderstanding to the readers while interpreting the data (Hardness depends on the grain size while fracture toughness shows independence).

  1. Letters (a, b, c) in figures 3 and 7 must be made the same.

Response: Thanks. We have edited the Figure. 3 and 7 to be in consistency.

  1. p.16, line 525. The value "... 0.02 ps-1" is better to indicate with an exponent for easier understanding.

Response: Thanks. We have edited the value in line 525 to be of an exponent in the revised manuscript (as line 552).

References

  1. Rendtel, A., B. Moessner, and K.A. Schwetz, Hardness and Hardness Determination in Silicon Carbide Materials, in Advances in Ceramic Armor: A Collection of Papers Presented at the 29th International Conference on Advanced Ceramics and Composites, January 23‐28, 2005, Cocoa Beach, Florida, Ceramic Engineering and Science Proceedings. 2005. p. 161-168.
  2. Rao, X., et al., Characterization of hardness, elastic modulus and fracture toughness of RB-SiC ceramics at elevated temperature by Vickers test. Materials Science and Engineering: A, 2019. 744: p. 426-435.
  3. Rahman, A., et al., Mechanical characterization of fine grained silicon carbide consolidated using polymer pyrolysis and spark plasma sintering. Ceramics International, 2014. 40(8, Part A): p. 12081-12091.
  4. Ghosh, D., et al., Dynamic Indentation Response of Fine-Grained Boron Carbide. Journal of the American Ceramic Society, 2007. 90(6): p. 1850-1857.
  5. Liao, F., et al., Superhard nanocrystalline silicon carbide films. Applied Physics Letters, 2005. 86(17): p. 171913.
  6. Han, Z., et al., Microstructure and mechanical properties of boron carbide thin films. Materials Letters, 2002. 57(4): p. 899-903.
  7. Domnich, V., et al., Nanoindentation and Raman spectroscopy studies of boron carbide single crystals. Applied Physics Letters, 2002. 81(20): p. 3783-3785.
